

# A low-maintenance optoacoustic sensor for black carbon monitoring

Linda Haedrich[1,2*], Nikolaos Kousias[3*], Ioannis Raptis[3], Uli Stahl[1,2], Leonidas Ntziachristos[3], Vasilis Ntziachristos[1,2,4]

[1]Chair of Biological Imaging, Central Institute for Translational Cancer Research (TranslaTUM), School of Medicine and Health & School of Computation, Information and Technology, Technical University of Munich, Munich, 81675, Germany
[2]Institute of Biological and Medical Imaging, Bioengineering Center, Helmholtz Zentrum München, Neuherberg, 85764, Germany
[3]Mechanical Engineering Department, Aristotle University of Thessaloniki, P.O. Box 458, GR 54124 Thessaloniki, Greece
[4]Munich Institute of Biomedical Engineering (MIBE), Technical University of Munich, Garching b. München, 85748, Germany
[*]These authors contributed equally to this work

*Correspondence to*: Vasilis Ntziachristos (bioimaging.translatum@tum.de)

**Abstract.** Regulation of black carbon (BC) emissions is necessary due to their negative impact to climate and human health. We present a low-cost optoacoustic sensor for Black Carbon (BC) emissions, which can provide continuous measurements that are suitable for long-term BC monitoring in highly contaminative environments with low need for frequent maintenance. Insensitivity to contamination is based on a sensor design that integrates protective flows of clean air around the sample measured, which minimizes BC deposition on the detector and optical windows of the sensor. A quantitative analysis shows that the negative effect of BC contamination on sensor performance is reduced by a factor of greater than 300 000 in comparison to an unprotected control sensor. We discuss how the reduced maintenance requirements make the design presented a promising candidate for continuous and long-term BC monitoring of high emitters, enabling disseminated monitoring necessary for regulatory and mitigation measures of BC emissions in the future.

## 1 Introduction

Black carbon (BC) particles are generated by pyrolysis of carbon-containing fuels, especially during incomplete combustion, and can have a significant impact on the climate and human health (Chowdhury et al., 2022). Two thirds of all BC emissions originate from anthropogenic activities (Bond et al., 2013; Timonen et al., 2019), while the rest are emitted by wildfires. The main anthropogenic emission sources of BC include residential wood burning, transportation and industry (Xu et al., 2021). In winter, BC emissions from wood burning can be up to 30% of all BC emissions for urban areas in Europe (Kalogridis et al., 2018). In regions of Asia and Africa where wood burning is used for cooking, the effect can be even higher. The concentration of BC at the exhaust of stoves was measured in rural China and reached concentrations of 50-80 mg/m$^3$ (Shen et al., 2020).

Road transport pollutant emissions, including BC, have been reduced effectively by the Euro standards (Union, 2009), but remain the main BC source in Europe. Shipping emissions are relatively low in terms of their total emissions ~1.7% of all BC



emissions (Lack et al., 2008) or ~2.9% of total emissions (IMO, 2021), but they have a significant climate warming effect when emitted within the Arctic Circle (Kang et al., 2020). Shipping also majorly affects air quality in cities, due to high local

emissions at ports. Regarding industry emissions, coke ovens and brick kilns are the largest emitter sectors (Xu et al., 2021). Coke ovens are mostly located in China. Recent efforts are aimed toward the use of cleaner fuel and the establishment of control technologies to limit BC emissions. Brick kilns remain a significant emitter of BC particles in Asia. For example, Bangladesh has ~5000 thousand brick kiln facilities. The BC concentration at the exhaust of a brick kiln can reach concentrations of 10-15 mg/m$^3$, and they operate for 5-6 months depending on monsoon season (Haque et al., 2018). It is

important to note that Bangladesh is very close to the Himalayas, a region that is affected disproportionally by BC emissions due to excessive snow coverage, similar to the Arctic.

BC yields two main negative outcomes: climate warming and negative health effects. The global climate warming effect of BC is calculated to be between +0.15W/m$^2$ (Arias et al., 2019) to +1.1 W/m$^2$ (Bond et al., 2013). The lower estimates do not

include the warming potential of organic aerosols, which has been shown to be on the order of +0.22 to +0.57 W/m$^2$ (Lin et al., 2014). The climatic effects of BC are much larger in snow-covered regions, due to accelerated snow melting and positive albedo feedback. Estimates of an additional warming effect of +0.17 W/m$^2$, +1.5 W/m$^2$ and up to more than 100 W/m$^2$ (Kang et al., 2020) have been reported for the Arctic, the Alps and the Tibetan Plateau, respectively. With regards to health effects, BC emissions relate to cardiovascular mortality and morbidity, particularly with very small particles (<300 nm) reaching the

bloodstream through breathing and the lungs, due to toxic substances that are co-emitted with BC and then absorbed by the carbon particles (Janssen et al., 2012). Additionally, BC can cause respiratory effects (Lepisto et al., 2022), and has recently been associated with brain tumours (Poulsen et al., 2020).

The reduction of the negative impact of BC requires legislation for BC emissions control, expected to positively impact climate

warming mitigation while providing health benefits for the general population (Brewer, 2019). BC has been included in the latest Air Quality Directive by the European Commission (Community, 2024), which includes mandatory ambient BC measurements where concentrations are expected to be large, e.g., in street canyons. Source specific legislation that includes direct exhaust measurements for selected sources and at selected locations where BC has increased warming potential (e.g. Arctic, Himalayas) would be most beneficial. The International Maritime Organization (IMO) has further made an effort to

establish protocols for on-board monitoring of BC from ships with a focus on emissions in the Arctic. The IMO has also suggested suitable measurement techniques, all of which rely on the optical absorption properties of BC (IMO, 2011, 2015, 2018). Still, the potential of current sensors is limited by either sensor cost or maintenance requirements, as there are no low-cost systems that can withstand the demanding environment of vessel funnels without frequent maintenance. The same limitations apply for other direct exhaust measurement applications, such as in industry. Such a low-cost sensor could

accelerate the introduction of legislation for direct BC monitoring from individual relevant sources, e.g., ships, brick kilns, cooking stoves, fireplaces, etc.



The main technologies that are used to quantify the concentration of BC particles are filter smoke numbers (FSN), aethalometers for ambient measurements, laser-induced incandescence (LII) and optoacoustic spectroscopy (OptAS), also often referred to as photoacoustic spectroscopy (PAS) (IMO, 2018; Aakko-Saksa et al., 2022). They all rely on the optical properties of BC to quantify its concentration. Smoke meters (SM), which determine the FSN, and their ambient-concentration counterparts known as aethalometers, detect BC particles that are deposited on a filter tape. SMs rely on the reflectance of the filter (Giechaskiel et al., 2014) while aethalometers rely on the attenuation of light (absorption plus scattering) through the filter (Hansen et al., 1984). Both technologies suffer from some key limitations. The most important one is scattering artefacts which are shown to reduce the sensor's accuracy (Kim et al., 2019), especially in the presence of particles with high scattering albedo. There is currently no correction for this artefact. Additionally, filter-based instruments suffer from the filter loading artefact, which can be compensated by using the dual spot method (Drinovec et al., 2015). However, this correction method is only implemented by lab-grade instruments. Lower cost systems rely on less accurate correction algorithms (Coen et al., 2010). For raw exhaust applications, condensation of water on the filter tape can be a significant issue and often a heated sample inlet is used to reduce humidity (Backman et al., 2017). Filter-based instruments operate optimally for ambient applications where the concentration of BC particles is small. Thus, using filter-based instruments for exhaust applications will lead to increased maintenance requirements, especially for miniaturized versions of the technology. For aethalometers, daily maintenance with a high cost of replacement filters and additional manpower adds to the retail prices of the instruments. FSN measurements are mainly used during engine certification and inspection and cannot be carried out in real-time, as time is needed for BC to accumulate on the filter paper (Aakko-Saksa et al., 2022).

Laser Induced Incandescence (LII) sensors use very high energy lasers to heat BC particles up to 4000 K so that they radiate in the visible spectrum, which is then measured by optical detectors (Michelsen et al., 2015). Due to the required high power, LII sensor systems are very expensive and have small potential for cost reduction and technology miniaturization. Thus, despite their favourable measurement characteristics, such devices are not suitable for widespread implementation.

OptAS sensors rely on light pulses to induce periodic energy absorption by BC particles which leads to the emission of pressure waves from the particles to the surrounding media. The generated acoustic signal is directly proportional to BC concentration and is detected by a sound transducer. It is common to use acoustic resonators to enhance the sound intensity and improve sensitivity (Ma, 2018). The most widely used commercial OptAS instruments are the AVL Micro Soot Sensor (MSS; AVL, Graz, Austria) (Schindler et al., 2004) and the Photoacoustic Extinctiometer (PAX; Droplet Measurement Technologies, CO, USA). The cost of these instruments is very high, making their widespread application challenging, similar to LII sensors. In addition, they employ resonators which make the system very sensitive to changes in environmental conditions and require detailed sample conditioning. Low-cost OptAS sensors for gas sensing have been developed by implementing a relatively new technique named Quartz Enhanced Optoacoustic Spectroscopy (QEPAS) (Patimisco et al., 2014). This technique relies on a



quartz tuning fork (QTF), an ultrasound transducer with a very high Q factor, which is simultaneously very sensitive and of minimal cost. Despite the success of QEPAS sensors for gas sensing, there is still a lack of particle sensors that rely on this technology. The limiting factor for the application of QEPAS for particle sensing is the deposition of particles on the QTF, which can potentially lead to its degradation and loss of sensitivity.

105

In summary, there is currently a lack of affordable sensors with low maintenance requirements, suitable for use in highly contaminative environments, such as the direct exhaust monitoring of high emitters of BC particles. In this work, we present the performance of a new sensor design, termed illumination-detection separating sensor (IDSS) (Stylogiannis et al., 2021; Ntziachristos et al., 2021), that enables direct exhaust measurements of high emitters such as ships, brick kilns and more without requiring maintenance for long periods of time. The design of the IDSS separates the sensitive element, a quartz tuning fork (QTF), from the sample flow containing BC particles and therefore direct contamination of the QTF is avoided, preventing sensitivity loss. To avoid diffusion of particles out of the sample flow, the sensor design integrates protective flows of clean air. The flows eliminate BC deposition on inward-facing optical windows for the excitation laser, which could otherwise result in the generation of an ever-increasing acoustic background signal.

115

The IDSS was already successfully implemented on-board the Stena Germanica, a RoPax ferry (roll-on-roll-off ferry transporting passengers and cargo) for continuous measurements of BC particles (Haedrich et al., 2025). Herein, we show tests from laboratory measurements with the IDSS at higher BC concentrations and for a longer duration than the previous field tests on-board the Stena Germanica. We show that the protective flows eliminate particle deposition onto the optical windows even at higher emission concentrations, resulting in a stable sensor performance. We further estimate that an optimized sensor could operate for at least 1.5 years on board a ship before maintenance is required.

## 2 Methods

### 2.1 Design and production of the IDSS and the control sensor

We previously describe the development, construction and design of the OptA IDSS in (Ntziachristos et al., 2021; Stylogiannis et al., 2021; Haedrich et al., 2025). To summarize, the IDSS consists of a 3D-printed ellipsoidal cavity with two focal points (FP1 and FP2) separated by a distance of ~50 mm. The sensing element, a quartz tuning fork (QTF; 100 kHz, Type TC-26, Conrad, Hirschau, Germany) is installed at one of the focal points. At the other focal point, the BC sample flow is irradiated by a laser excitation, generating the acoustic signal. By design, the ellipsoidal cavity focuses the propagating acoustic signal onto the QTF.

We incorporated particle-free airflow around the sample flow to prevent the diffusion of particles into the ellipsoid chamber containing the QTF and onto the laser optical windows. Additional flows leading away from the optical windows added extra protection against particle deposition onto optical surfaces. Protective flows were integrated into the IDSS through spaces



integrated into the design of the 3D-printed resin comprising the main body of the IDSS. A control sensor without contamination protection was 3D-printed without these spaces integrated into the sensor design. Both the IDSS and the control sensor were printed using a Form 2 printer (Formlabs GmbH, Berlin, Germany).

## 2.2 Assessment of the IDSS performance in the laboratory

We performed laboratory tests to assess the performance of the IDSS compared to (1) an unprotected control and a reference MSS (AVL, Graz, Austria) after exposure to 0.046 mg BC (Fig. 1a), and (2) an MSS after exposure to 7.8 mg BC (Fig. 1b). For Test 1 (0.046 mg BC exposure), the devices tested measured the same BC sample in parallel (Fig. 4a). A propane-fuelled Argonaut burner (Miniature Inverted Soot Generator, Argonaut, Edmonton, Canada) was used to generate BC particles. To control the BC concentration, the sample was diluted with a Dekati Fine Particle Sampler (FPS, Dekati, Kangasala, Finland) and the sample was provided to both OptA sensors with a flowrate of 1.5 lpm each and to the MSS with a flowrate of 2 lpm. The final sample flow rate for the IDSS was 2.2 lpm, as it included clean air of 0.7 lpm total protective flows. The MSS operated with additional internal dilution (DR = 1:5), which reduced the exposure of the MSS to BC compared to the IDSS and its control.

For Test 2 (7.8 mg BC exposure), the IDSS was stress tested to evaluate its performance compared to the MSS over extended periods of high BC mass exposure (Fig. 1b). Two burners were used during the stress test to produce BC emissions. BC particles for the first half of the test were generated using a propane-fuelled APG burner (Aerosol Particle Generator, AVL, Graz, Austria). The APG (Fig. 1) has internal dilution, so no external diluter was necessary. The second half of the stress test was performed with the Argonaut burner as described in Test 1. The same flow configuration was maintained for the IDSS in this experiment (i.e., 1.5 lpm sample flow, 0.7 lpm protective flow). An MSS was again used as a reference instrument for measuring BC concentration. The MSS operated with an additional DR equal to 1:20 during this test, which reduced its exposure to BC particles. This means that the MSS was subjected to a lower BC mass concentration than the OptA IDSS.





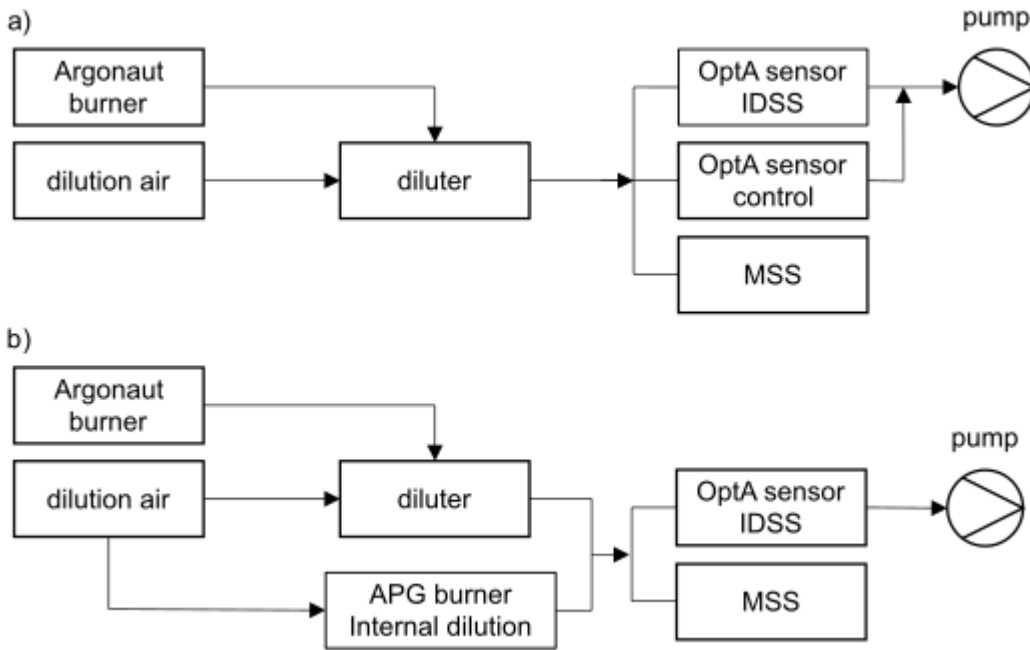

**Figure 1: Schematic of the laboratory setups for sensor performance assessment.** a) Schematic of the setup to test the illumination-detection separating sensor (IDSS) in comparison to the unprotected control optoacoustic (OptA) sensor during exposure to the same sample in parallel. An AVL Micro Soot Sensor (MSS) is used as reference. b) Schematic of the setup used to stress test the IDSS at high levels of black carbon (BC) exposure. An AVL MSS was used as a reference. APG: Aerosol Particle Generator.

## 2.3 Baseline correction

Baseline correction was necessary for data collected by the unprotected control sensor to separate actual BC measurements from contamination-induced background signals. By fitting a line through the flushing periods, we can estimate the baseline increase caused by contamination during the exposure periods. During the exposure periods it is not possible to fit through the data as we cannot distinguish between signal from contamination and signal from the sample. The baseline was calculated in two steps. First, we calculated linear fits for each individual flushing period. Second, linear fits during the exposure cycles were applied such that the baseline before each exposure cycle was connected to the baseline after the exposure cycle.

## 2.4 OptA signal conversion to BC concentration

For the unprotected control sensor, we first subtracted the fitted baseline from the raw data as described above. Next, we plotted baseline-subtracted OptA data from the control sensor and the raw data of the IDSS against measured BC concentrations from the MSS, and found a linear relationship between each prototype sensor and the MSS (Supplemental Fig. 1a,b). We used this linear relationship to convert OptA data points to BC concentrations.





The calculated BC concentrations for the IDSS and the control were then compared to the BC concentrations measured by the MSS to quantify deviations from the reference instrument. We calculated the ARD and the MARD of the exposure periods using Equation 1, where $c_{OptA}$ is the calculated BC concentration from the OptA sensor and $c_{ref}$ is the BC concentration measured by the reference instrument.

$$MARD = mean(ARD) = mean(abs(c_{OptA} - c_{ref})/c_{ref}) , \qquad (1)$$

### 2.5 Calculation of maintenance periods

We can calculate the time between maintenance periods (t) in days using Equation 2 below,

$$t = \frac{m_{BC}}{C_f \, t_{hours} \, FR \cdot c_{BC} \cdot DR} \qquad (2)$$

where $t_{hours}$ refers to the ship's operational hours per day, $m_{BC}$ refers to the maximum BC mass in mg the sensor should be exposed to between maintenance periods, DR refers to the sample dilution ratio, $c_{BC}$ refers to the undiluted BC concentration emitted by the vessel in mg/m$^3$, FR refers to the flow rate through the sensor in lpm and $C_f$ is a unit conversion factor equal to 0.06.

### 2.6 Kolmogorov-Smirnov test

To check if the IDSS (raw) performs differently than the unprotected control (after baseline subtraction), a two-sample Kolmogorov-Smirnov test was performed. With this test we checked whether the ARD of the IDSS follows the same continuous ARD distribution as the control sensor. The tests were performed in Matlab R2022b (MathWorks, MA, USA).

### 3. Results

In the ellipsoidal cavity of the IDSS, the QTF is separated from the BC sample flow by a distance of ~50 mm. This separation has been observed to practically eliminate particle deposition onto the QTF itself (Haedrich et al., 2025). A protective sheath flow of clean air has been integrated into the sensor (Fig. 2a), to avoid the contamination of the sensor volume because of BC particles diffusing out of the sample flow. A second sensor with the sheath flow switched off was used as a control in order to examine the effectiveness of the sheath flow. The OptA signal recorded by the control sensor is depicted in Fig. 2b, where it is evident that the baseline signal increases with measurement time as the sensor is exposed to BC particles.





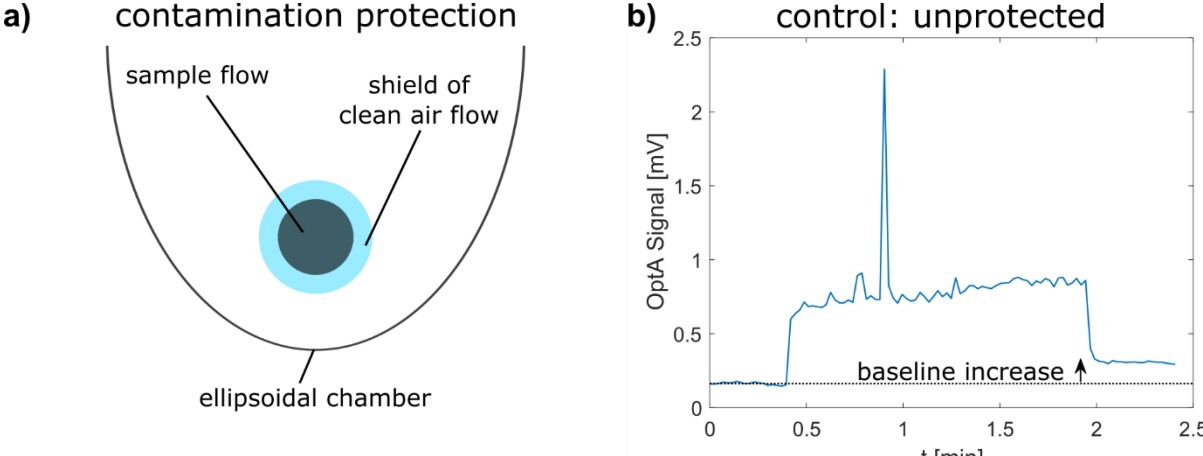

**Figure 2: Sketch of the sheath flow integrated into the illumination-detection separating sensor (IDSS) that prevents a baseline increase in optoacoustic (OptA) signal.** a) Cross-section of the ellipsoid cavity of the IDSS sketched to depict the sheath flow of clean air surrounding the sample flow containing black carbon (BC) particles. b) Representative sample of the OptA signal recorded over 2 min by a control sensor with the sheath flows turned off. The graph shows a clear baseline increase, caused by contamination of the sensor.

The impact of the sheath flow on OptA signals was quantified through an experiment wherein the IDSS, operating normally, and the control sensor were exposed to BC over a long interval, with intermediate flushing periods to monitor the baseline. During this experiment, each sensor was exposed to a total BC mass of 45.7 µg. This mass corresponds to an exposure time of 38 h at a flowrate of 2 lpm and a BC concentration of 10 µg/m³. A reference OptAS instrument, an AVL MSS, was in operation at the same time, but this instrument was exposed to fewer BC particles as it had an additional 5-fold internal dilution system and a different sample flowrate, resulting in a final exposure of 12.7 µg BC. Figure 3a depicts the results of this experiment for all three instruments. The exposure periods and intermediate flushing periods are clearly visible. Raw, unprocessed data from the IDSS and the control sensor are presented in mV to showcase the baseline increase for the control sensor and the lack of baseline increase for the IDSS. The baseline OptA signal obtained from the control sensor increases with each exposure step, while the IDSS's baseline remains stable. Due to the baseline increase, it was not possible to convert the raw data of the control sensor to BC concentration.

We investigated whether the contamination effect of the control sensor could be corrected during post-processing using baseline subtraction. This correction method accounts for the baseline increase during the exposure periods by subtracting a linear fit, which connects the baseline before and after each exposure interval (see Methods). The corrected data from the control sensor is plotted alongside data from the other sensors in Fig. 3a, overlapping with the uncorrected data from the IDSS. For better visibility, a zoomed-in view of the last exposure cycle is plotted in Fig. 3b.



The zoomed-in view (Fig. 3b) shows that the data collected by the IDSS are more stable compared to the corrected data from the control sensor. Here, the OptA data obtained from both sensors were converted to BC mass concentrations using a linear

relationship derived from the OptA sensor data and the BC concentrations obtained by the MSS (Supplemental Fig. 1a, b). The conversion factors used to convert the OptA signal in mV to BC concentration in mg/m³ were $0.842 \pm 0.007$ mg/m³ per mV for the baseline-subtracted data from the control and $1.043 \pm 0.002$ mg/m³ per mV for the raw data of the IDSS. Importantly, the signal measured by the IDSS exhibited a similar trend as the expensive AVL MSS.

To investigate deviations in the signal between the OptA sensor and the MSS, we calculated the mean absolute relative difference (MARD) of both the IDSS and control sensors, using the AVL MSS as a benchmark. Figure 3c shows both the MARD and the absolute relative difference (ARD) of each prototype sensor calculated from the data plotted in Fig. 3a. For the control sensor, we calculated a MARD of $22.26 \pm 27.17\%$, compared to a MARD of $4.54 \pm 5.76\%$ for the IDSS. This confirms the superior performance of the IDSS compared to the unprotected control sensor. A two-sample Kolmogorov-

Smirnov test shows that the ARD of the IDSS has a significantly different distribution to that of the control ($p = 0.001$).

We further investigated the background noise for both the IDSS and the control sensor, which we defined as the standard deviation (SD) of the OptA signal recorded during the flushing periods. Figure 3d shows that the background noise of the unprotected control sensor increased with BC exposure, while the IDSS's background noise stayed low throughout the

measurement period. After only 6 exposure cycles, the mean SD of the control sensor was around 12 times larger than that of the IDSS. The IDSS's background noise levels followed a similar trend as the AVL MSS, suggesting that the high SD of the control sensor was caused by the presence of small amounts of BC particles during the clean air periods. The increase in the background noise (compared to a baseline increase only) makes it ineffective to correct the OptA signal with a simple background subtraction. Overall, Fig. 3d shows that the sheath flow greatly improves the performance of the sensor.






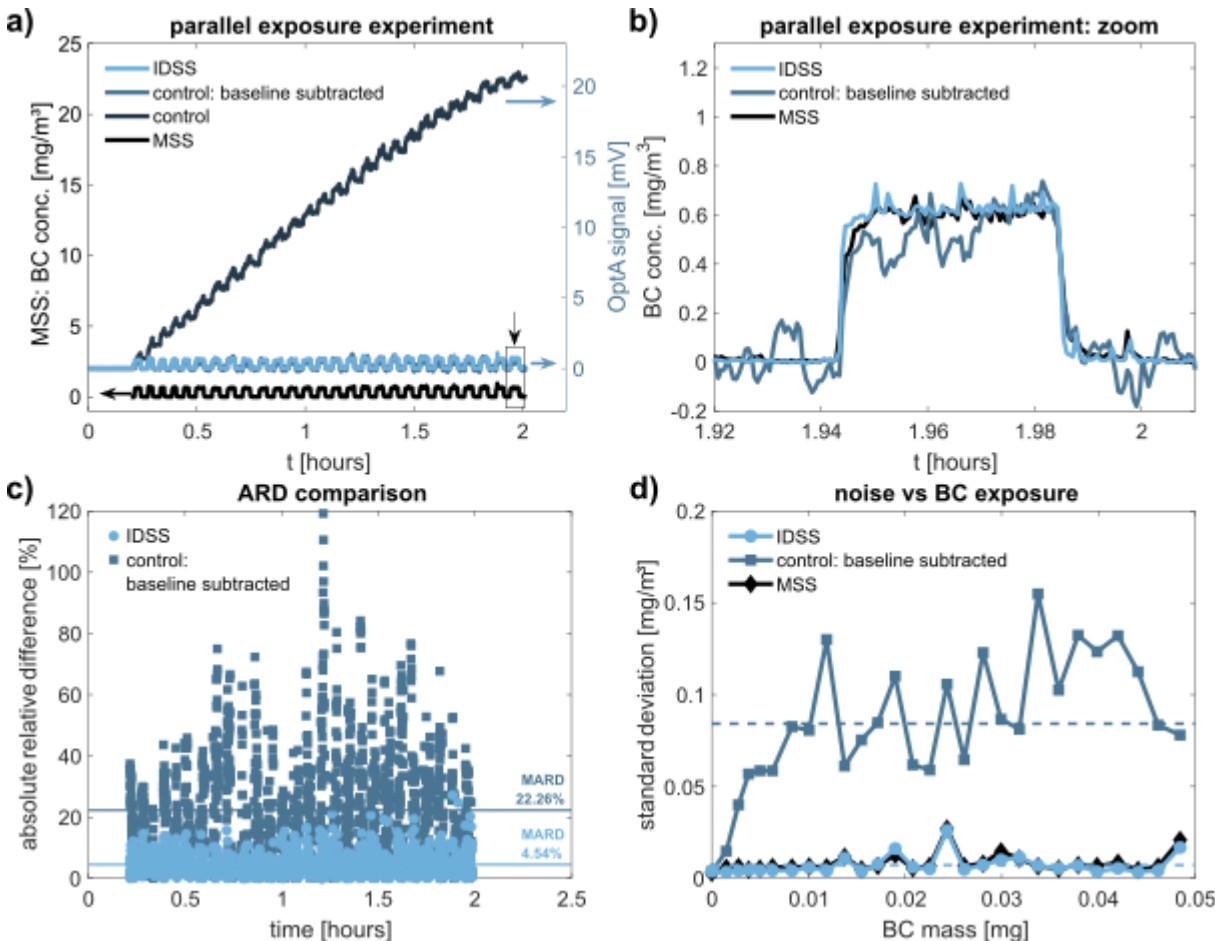

**Figure 3: Performance comparison of the illumination-detection separating sensor (IDSS) to a control sensor under prolonged exposure to black carbon (BC). a) The signal of the IDSS (raw) and the control (raw and baseline-subtracted) during exposure to 46 µg BC. An AVL Micro Soot Sensor (MSS) was used as a reference instrument (exposed to 12.7 µg BC). b) A detailed view of the last measurement cycle, indicated by the black box and arrow in (a). c) The absolute relative difference (ARD) and the mean absolute relative difference (MARD) of the IDSS (raw) and the control (baseline-subtracted) in relation to the MSS. d) The standard deviation (SD) of the background noise for each exposure period plotted versus the respective cumulative BC mass exposure of each sensor.**

We conducted a stress test of the IDSS to examine its performance when sampling high quantities of BC over a measurement period. Briefly, we exposed the IDSS to a total of 7.8 mg BC, ~171-times more BC than in the previous experiment. The MSS measured the same BC sample in parallel but was exposed to only 0.52 mg BC due to the 20-fold additional dilution done by the MSS itself. Figure 4a shows raw data from the IDSS converted to BC concentration via the linear relationship presented in Supplemental Fig. 1b and the BC concentration measured by the MSS in the last few hours of the experiment. The two sensors exhibited similar trends; however, the MSS did not capture the magnitude of the rapidly changing spikes (fast changes in BC concentration) to the same extent as the IDSS. This discrepancy was likely due to the MSS's additional internal dilution system and higher sample volume, which resulted in concentration averaging over time within the MSS measuring cell. After



the high BC exposure experiment, we opened the IDSS to visually examine the ellipsoidal cavity, flow inlet and outlet (Supplemental Fig. 2). We only observed visible particle deposition on the sample inlet. The rest of the cavity appeared clean, suggesting that the sheath flow successfully encapsulates the sample flow and prevents particle diffusion out of the sample

flow.

Figure 4b shows that there is no visible increase in the background signal of the IDSS, even after high BC mass exposure to 7.8 mg BC, whereas the background signal of the control reached 21 mV by the time measurements were terminated, after sampling only 45.7 µg BC. A closer examination of the dataset from the control sensor reveals that the curve starts flattening

from 0.01 mg (Fig. 4b inset). This is due to a saturation effect, likely caused either by fewer and fewer BC particles being able to attach as the optical surfaces get covered with previously deposited BC particles or by BC particles attaching in the "shadow" of already deposited particles, i.e., these particles are not excited as the laser excitation is blocked by the previously attached particles. We applied linear fits (red lines) to the data points from the control sensor (up to 0.01 mg BC) and the IDSS (Fig. 4b). We found that the IDSS's trend line had a slope $3 \times 10^5$-fold lower than the control's trend line, meaning that the sheath flow

increases the lifespan of the IDSS (without cleaning) by a factor of $3 \times 10^5$ compared to the control sensor.

To put it into perspective, if we continued with the stress test as shown in Fig. 4a, the IDSS would be able to sample 511 mg of BC over a period of ~2000 hours before having the same amount of background signal as the control sensor after the first measurement cycle, i.e., 0.87 mV after 2 minutes (Fig. 3b inset). We want to note that the $R^2$ value of the fit for the IDSS is

relatively low ($R^2 = 0.05$). This indicates that there is no strong linear relationship between the background signals of the IDSS and the increasing amount of BC sampled. Thus, drifts in the instrumental noise over time might have caused a change in the baseline with the actual rate of contamination being even lower. However, we will use the slope of 0.0017 mV/mg baseline increase for the operation time estimations of the IDSS in real-world implementation scenarios.

From Fig. 3d, we can estimate that the noise, i.e., the SD, would increase by a factor of ~2 for this baseline increase of 0.87 mV resulting in a 2-fold decrease in sensitivity. For the following operation time estimations, we used the stringent value of 0.87 mV as the limit in background increase before cleaning the sensor is recommended. We will use the background signal increase of 0.87 mV (which corresponds to 511 mg of BC exposure) and the contamination rate of 0.0017 mV/mg to extrapolate the maintenance-free operation times to a few use cases to demonstrate the IDSS's capabilities. The mean BC concentration

measured on-board the Stena Germanica was 4.5 mg/m³ without dilution (see introduction). Continuous sampling at 2 lpm for 24 h/day without dilution means that 511 mg of BC will be sampled in 1.7 months (blue line in Fig. 4c). It is important to note that this use case depicts an extreme scenario for long continuous shipping routes. Furthermore, direct exhaust measurement is unlikely to occur without dilution. The time to cleaning or maintaining the sensor can be further prolonged by increasing the dilution ratio (DR) and decreasing the flowrate through the sensor (see methods), while also considering that ships do not

necessarily operate 24 h/ day.



With a DR of 1:150, a flowrate of 0.5 lpm and the same 16 h/day operation as the Stena Germanica (green line in Fig. 4c), 511 mg of BC sampled will theoretically be reached in 97.2 years. This means that maintenance and checking of the device can be safely done every 1.5 years (corresponding to the tested 7.8 mg BC exposure) without worrying about decreased

performance due to BC contamination, making this sensor a forerunner in low maintenance sensors. We believe this is a manageable cleaning schedule that results in negligible contamination of the optical windows, eliminating the need for baseline corrections.

**Figure 4: Stress testing the illumination-detection separating sensor (IDSS) under high black carbon (BC) mass exposure.** a) The data captured during the last hours of a BC exposure stress test of the IDSS (blue). The AVL Micro Soot Sensor (MSS, red) measured the





same sample in parallel and was used as a reference instrument. b) Background signal values (blue) of the unprotected control from the
dataset depicted in Fig. 3a and from the IDSS (panel a) plotted versus the total BC mass measured by the sensors. Linear fits (red) were
applied to quantify the rate [mV/mg] at which the optoacoustic (OptA) signal increases with BC exposure. c) Projection of BC mass sampled
by the IDSS over time for two use cases using the calculated slope from (b), which is 0.0017 mV/mg. DR: dilution ratio, FR/lpm: flowrate
in litres per minute, hrs/day: ship operation hours per day, e.g., the Stena Germanica operates around 16 hours/day.

## 4. Discussion

In this paper, we demonstrate that the IDSS operates with high stability even under extremely high BC particle mass exposure.
The high stability of the sensor is achieved by encapsulating the sample flow with a sheath flow, eliminating the diffusion of
BC particles out of the sample flow. This makes the sensor a promising candidate for low-maintenance BC monitoring in high
emission conditions, particularly shipboard environments.

Particle contamination is a challenge for ambient or exhaust measurements across all BC sensors, not only low-cost sensors.
The development of low-maintenance sensors is critical in order to enable the enforcement of BC emission-limiting regulations
and to reduce the environmental and health impacts of BC. Particle deposition and contamination, especially on the optical
windows, has been reported to increase background noise in OptA gas sensing applications (Miklós et al., 2001) and in the
MSS (Schindler et al., 2004). The MSS has successfully reduced the signal increase caused by contamination by integrating a
counter flow into the system, directing BC particles away from both optical windows (Schindler et al., 2004). Similarly, we
found that preventing BC deposition on the IDSS's optical surfaces almost eliminated the increase in background signal and
noise observed in a control sensor with the sheath flow switched off.

The incorporation of sheath flows essentially minimizes our sensor's maintenance requirements, avoiding the need for frequent
cleaning or replacement of the optical windows. As a result, our sensor can easily be used without cleaning maintenance for
1.5 years in various applications, such as on-board ships with the appropriate sample flow and DR. The extended maintenance
cycle makes the integration of the IDSS in highly contaminative environments suitable and cost effective.

In comparison, currently available sensor systems have much shorter maintenance cycles. For example, aethalometers require
frequent filter replacements. During the campaign on-board the Stena Germanica (Haedrich et al., 2025) the filters of the
ObservAir had to be replaced daily as indicated by the instrument. Filters for this type of sensors can be expensive, resulting
in high maintenance costs for long-term applications in highly contaminative environments. The need to involve trained
personnel to perform daily maintenance on the sensor significantly increases these maintenance costs. However, even lab-
grade aethalometers used to measure low BC concentrations require relatively frequent maintenance. In a recent study, the
filter rolls of aethalometers used to measure only ambient BC concentrations had to be exchanged 2-3 times per year (Mendoza
et al., 2024), meaning that they would need to be exchanged even more often when measuring high BC concentrations. Even
without filters, current OptAS instruments also have high maintenance requirements. For example, monthly cleaning of the



MSS is recommended, even with its integrated counter flow (Schindler et al., 2004). Thus, compared to currently available sensor systems, our sensor has the lowest maintenance requirements. These low maintenance requirements would enable our sensor to be integrated economically in environments with high BC concentrations, for example, on-board a ship without disrupting regular shipboard operations.

In this paper, we conservatively predicted that our sensor would be able to operate without maintenance for 1.5 years on ships similar to the Stena Germanica, which operates up to 16 hours a day. Even the longest global shipping routes rarely require more than ~45 days of consecutive travel at a time (Kuroda and Sugimoto, 2022; Dr. Theo Notteboom, 2020-24), well within our recommended 1.5-year interval between maintenance periods. Therefore, we expect that our sensor can be applied to all types of ships with no maintenance needs while ships are underway. Also, it is worth noting that the proposed 1.5-year maintenance interval is a conservative recommendation based on a slight increase in OptA background signal. At this level of background signal increase, our sensor still operates well with baseline correction. In fact, our stress test failed to find the upper limit of mass exposure at which baseline correction would no longer be sufficient, suggesting that the sensor could be operated with longer maintenance cycles or on ships with higher BC emissions at lower DRs.

As mentioned before, a significant amount of BC emissions come from brick kilns that have BC concentrations of 10-15 mg/m$^3$ (Haque et al., 2018), around 3 times higher than the average concentration assumed in our estimations for the Stena Germanica. In such environments, our sensor could conservatively operate for at least 0.5 years without maintenance, assuming the same DR and operating hours. Since monsoon season limits the operation period of these brick kilns to half a year, this is an acceptable timeframe for maintenance cycles. Biofuel kitchen stoves in rural China reach BC emission concentrations of 50-80 mg/m$^3$ (Shen et al., 2020), the highest reported BC concentrations emitted globally. However, these stoves operate 2-4 times per day for 10-30 minutes, with peak BC emissions occurring for only 5-10 minutes during fuel ignition stages (Shen et al., 2010). Assuming mean concentrations of 30 mg/m$^3$ for 1-2 hours a day, we estimate that our IDSS sensor could be implemented in such environments with a maintenance cycle of 0.8 – 1.5 years.

In summary, we present a novel OptA sensor capable of continuously measuring BC long-term in highly contaminative environments with minimal maintenance requirements. This capability is directly related to the sensor's design, which enables BC measurement without directly exposing the sensitive elements to the sample flow. The acoustic signal is transmitted through an ellipsoidal cavity allowing spatial separation of BC particles and the QTF. Furthermore, integrated protective air flows prevent particle deposition on the laser optical windows. Our sensor's low maintenance requirements make it a promising candidate for widespread implementation on-board ships and in other high-emission locations, where it could be used to monitor BC and facilitate establishment and enforcement of regulations.



## Data Availability

The sensor data used for creating the figures can be downloaded from Zenodo (https://doi.org/10.5281/zenodo.17190856).

## Author Contributions

All authors contributed to designing the IDSS and devising the study. NK and IR performed the measurements. LH analyzed the data. LH and NK prepared the first version of the manuscript. All edited the manuscript into its final submitted form.

## Competing interests

V.N. is a founder and equity owner of Maurus OY, sThesis GmbH, iThera Medical GmbH, Spear UG, and I3 Inc. Authors L.N., V.N., U.S., L.H., N.K. and I.R. are inventors on a pending patent application (Application No. GR2025000576) related to the sensor described in this work.

## Acknowledgments

This project has received funding from the European Union's Horizon 2020 research and innovation programme under grant agreement No 862811 (RSENSE) and No 814893 (SCIPPER). We thank Dr. Serene Lee and Dr. Elisa Bonnin for their attentive reading and improvements of the manuscript.



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
