# Peer review of "A low-maintenance optoacoustic sensor for black carbon monitoring"

_EGUsphere, 2025_

## Referee Comment (RC2)

**Review of "A low-maintenance optoacoustic sensor for black carbon monitoring" by Haedrich et al.**

**General comments:**

In my opinion, the amount of contents and new achievements presented in this manuscript are both not rich enough to be published as an independent research paper. These results could be included in supplemental information or methodology section of one of other papers focused on applications (i.e., real-world observation) using this instrument.

**Specific comments:**

Page7 Line 174:
Define the acronyms ARD and MARD.

Page 7 Equations (1) and (2):
Physical quantities (i.e., concentration, time) should be written in italic style, while the other quantities or descriptors (i.e., BC, hours) should be written in regular style.

Page 7 section 2.6:
This very short section should be merged with one of the other sections.

Figures 3 and 4:
Resolution of these figures is too low for publication. Please follow the guidelines for Figure preparation.